# A Low-Cost Early Warning Method for Infectious Diseases with Asymptomatic Carriers

**DOI:** 10.3390/healthcare12040469

**Published:** 2024-02-13

**Authors:** Mauro Gaspari

**Affiliations:** Department of Computer Science and Engineering, University of Bologna, 40126 Bologna, Italy; mauro.gaspari@unibo.it; Tel.: +39-051-2094875

**Keywords:** COVID-19, test positivity rate, early warning methods, health system management

## Abstract

At the beginning of 2023, the Italian former prime minister, the former health minister and 17 others including the current president of the Lombardy region were placed under investigation on suspicion of aggravated culpable epidemic in connection with the government’s response at the start of the COVID-19 pandemic. The charges revolve around the failure by authorities to take adequate measures to prevent the spread of the virus in the Bergamo area, which experienced a significant excess of deaths during the initial outbreak. The aim of this paper is to analyse the pandemic data of Italy and the Lombardy region in the first 10 days of the pandemic, spanning from the 24th of February 2020 to the 4th of March 2020. The objective is to determine whether the use of early warning indicators could have facilitated the identification of a critical increase in infections. This identification, in turn, would have enabled the timely formulation of strategies for pandemic containment, thereby reducing the number of deaths. In conclusion, to translate our findings into practical guidelines, we propose a low-cost early warning method for infectious respiratory diseases with asymptomatic carriers.

## 1. Introduction

In the last three years, many scientists dedicated their efforts to the analysis of pandemic data. The severity of COVID-19 infections and the intensive use of hospital resources have been the driving factors behind these studies. One of the goals of these research efforts is to develop updated pandemic plans that provide alert indicators. These indicators enable the early detection of critical situations [1,2,3]. A critical aspect of the COVID-19 pandemic is the presence of a large number of asymptomatic cases, contributing to the rapid worldwide spread [4,5], presenting numerous challenges. Indeed, transmission due to asymptomatic infections cannot be effectively identified, making it more difficult to contain and control [6,7,8,9]. New and complex mathematical models are needed to represent epidemic dynamics due to asymptomatic carriers [10,11,12,13,14,15], also considering infection transmission prior to symptom onset [16]. Most importantly, early detection of aberrations becomes a crucial point [1,17].

At the end of February 2020, the first case was detected in Italy, and the virus began to spread rapidly, especially in the northern regions of Italy. Despite a national lockdown being enforced throughout Italy on the 9th of March 2020, just 15 days after the first case was discovered, three years later, in March 2023, the Italian former prime minister, the former health minister and 17 others including the current president of the Lombardy region were placed under investigation on suspicion of aggravated culpable epidemic. This investigation was conducted in connection with an alleged delayed government response and the high number of deaths in the Bergamo area, which registered a significant excess of deaths during the initial outbreak of the virus [18,19,20,21,22].

Considering the incidence rate, more precisely the 7-day incidence rate per 100,000 inhabitants, an indicator based on known cases that was used in successive waves for surveillance and establishing thresholds to identify critical areas, nothing conclusive can be drawn in connection to the beginning of March 2020. Indeed, analysis of the trend of incidence rate reveals that the values found on 9 March 2020 for Italy (11.88) and the Lombardy region (42.07) were both well below the “red zone” alert threshold (250), and even lower than the “white zone” alert threshold (50). The latter was used as a first warning threshold, not associated with drastic measures.

However, there are other indicators that could have been used at the initial stages. For example, the Test Positivity Rate (TPR), which is the percentage of positive tests over total tests [23], proved to be more suitable for modelling under-ascertainment due to asymptomatic carriers [24,25,26,27,28]. This indicator was demonstrated to be correlated with excess deaths in the first wave of the pandemic [29], and, most importantly, it anticipates incidence [23].

The aim of this paper is to analyse pandemic data for Italy and the Lombardy region in the first 10 days of the pandemic, spanning from the 24th of February 2020 to the 4th of March 2020. The goal is to determine whether the use of early warning indicators, such as TPR, would have made it possible to identify critical increases in infections, thus enabling the timely formulation of strategies for pandemic containment, reducing the number of deaths.

Finally, to translate these observations into practical guidelines, we propose a low-cost early warning method for infectious respiratory diseases with asymptomatic carriers.

## 2. Methods

The goal of early warning methods is to identify abnormal distribution of infectious diseases that may have the potential to develop into outbreaks with significant fatalities [3]. We analyse the first 10 days of the pandemic in Italy and Lombardy to identify a set of conditions in public health records that could be used to define an early warning method suitable for infectious diseases with asymptomatic carriers.

Making valuable predictions based on such a short time interval and limited data is a challenging task. For example, useful predictions based on positive cases and ICU admissions, concerning the potential strain on ICU beds in Italy, were developed approximately 10 days after the 4th of March 2020 [30]. The presence of asymptomatic carries makes the task even more challenging due to under-ascertainment issues in indicators based on known cases, such as the incidence rate.

In our analysis, we consider test positivity and community-level indicators to model the impact of COVID-19 on hospitalization, aiming to capture significant variations within a 7-day interval to generate a warning signal. Our objective is to answer the following question: is it possible to define an early warning method within a period of one weak at the beginning of a pandemic that correlates with excess mortality?

### 2.1. Modelling Test Positivity

Following the approach of [23], we use a definition of TPR based on two levels: one level deals with data collection issues, and the other with epidemiological issues. We let $t_{1}$, $t_{2}$, *…* $t_{n}$ be the daily TPR time series. TPR at time *i*, when i>3 and i<n−3, can be modelled by computing the trend as follows:(1)$${\bar{t}}_{i}=\frac{t_{i-3}+\ldots +t_{i+3}}{7}.$$

Using this approach, TPR on a given day is modelled by computing the average value of the days preceding and following it. Then, we introduce a second level to compute the final TPR value on Day *d* considering epidemiological issues. The TPR value is defined as the average of the last μ days of the previous time series, where μ is the incubation period.
(2)$$\tau^{d}=\frac{\left({\bar{t}}_{d}+{\bar{t}}_{d-1}\ldots {\bar{t}}_{d-\mu }\right)}{\mu },$$
where μ=5 on the onset [31], and μ=3 in the Omicron outbreak [32,33]. The details of the calculation are presented in [23].

A comparison between incidence rate and TPR in the first 10 days of the pandemic is presented in Figure 1 considering whole Italy (a) and Lombardy (b). While the values of incidence rate are relatively low in the first 10 days of the pandemic, TPR rises to 15% for both Italy (16.29%) and Lombardy (16.47%), not far from the 20% threshold that World Health Organization sets for the highest level of community transmission [34].

The low values for the incidence rate are, of course, related to the limited volume of tests conducted during those days; however, this is a typical condition at the beginning of an outbreak.

Would a significant increase in test positivity be a sufficient signal to justify the early enactment of drastic measures, such as the lockdown of the 9th of March 2020? Although a relationship between test positivity and mortality was discovered in the first COVID-19 outbreak in Italy [29], the analysis of pandemic data in the successive waves shows that an increase in test positivity alone is not a sufficient condition to identify emergency situations [23]. For example, a similar increase was observed in the Omicron outbreak at the end of December 2021, without the need to make equally drastic decisions. The point is that community-level indicators [35], which measure the impact of COVID-19 in terms of hospitalizations and strain on the healthcare system, should be also considered when making a decision.

### 2.2. Modelling Admissions in Hospitals and Intensive Care Units

A possible alarm signal could be associated with an estimate of the number of potential deaths, which was only 28 in the first 10 days for the entirety of Italy. Such an estimate can be obtained by analysing the trends in positive cases and hospitalizations during those days. Figure 2 presents the number of positive cases, the number of patients admitted in hospitals and ICUs in the first 10 days in Italy and Lombardy. While all the indicators demonstrate growth, the number of cases was still insufficient to deduce a critical situation during those days.

Analysing COVID-19 data from February 2020 to December 2023, we observed that the growth rates of positive and hospitalized patients were higher in the first few days than in the rest of the pandemic, as shown in Figure 3. This figure presents the growth rate trends of positive cases and hospitalized patients in Italy throughout the pandemic; a similar trend can be observed in all Italian regions. To delve into this point, we investigated whether high growth rates of positive cases and admissions to hospitals correlate with excess mortality. We considered average growth rates in 7-day intervals, observing that the regions where growth rates were higher than 40% included Lazio, Puglia and Toscana where the mortality rate was not particularly high. We also found that restricting the analysis to hospitals and ICUs only did not improve the results. For example, considering a growth rate threshold of 45% for admission in hospitals and ICUs, Lazio and Puglia were still included, while Lombardy, the first region of Italy considering mortality rate, was not.

In light of these considerations, we argue that growth rates of positive and hospitalized cases are not precise indicators for the purpose of defining an early warning method capable of detecting the risk of high mortality rates. On the contrary, we discovered two community-level indicators that appear to be more precise and suitable for our aims:The N hospital/positive ratio (HPRN), which is defined as the ratio between hospitalized patients and the number of positive cases N days before.The N ICU/hospital ratio (IHRN), which is defined as the ratio between patients admitted in ICU and hospitalized patients N days before.

N varies depending on the disease; for example, 4 could be a reasonable number for COVID-19 [36]. However, posing N equal to 4 generates a delay of 4 days in the computation of these indicators which is not compatible with the objective of providing an aberration signal in a week. Moreover, the value of 4, which is specific to COVID-19, may not be adequate for other diseases. However, we found that posing N equal to 1 still preserves both indicator properties, which instead are lost posing N equal to 0. The trends of HPR1 and IHR1 in Italy and Lombardy in the first days of the pandemic are presented in Figure 4, while the trends of these indicators throughout the pandemic in Italy are presented in Figure 5.

### 2.3. Defining an Early Warning Method

Combining these two indicators with test positivity, an early warning method can be formulated using the following rules of thumb which trigger warning signals in a 7-day interval:
IF TPR grows more than 5 points    AND the TPR is monotone    AND the growth rate of the HPR1 is greather than 5    AND the growth rate of the IHR1 is greather than 5    AND The average value of IHR1 is greather than 15THEN an aberration signal is detected IF the number of patients admitted in ICU is 0 for one day in the week.
    AND TPR grows more than 8 points    AND the TPR is monotone    AND The average value of HPR1 is greather than 20    AND The average value of IHR1 is greather than 10THEN an aberration signal is detected

The monotonicity condition on TPR is added to account for periods in which TPR effectively grows, excluding weeks that could contain possible peaks. The minimum threshold on the average value of the IHR1 allows for us to discriminate critical situations. Indeed, after the spread of the Omicron variant at the end of December 2021, the IHR1 dropped by half [23], reducing healthcare system strain.

The second rule addresses situations in which data on ICU are missing in the first days. This was the case of three Italian regions: Valle d’Aosta, Piemonte and Alto Adige. For these regions, the first rule is not triggered because the data on ICU are missing, namely the number of accesses in ICU is 0, and thus the two conditions that use growth rates do not hold simultaneously. To resolve this issue, we relax the condition on the average value of the IHR1, making other conditions more selective. However, the reported problem could be also related to delays in reporting data, a crucial issue for early warning systems and surveillance in general [37].

## 3. Results

Using the above early warning method, an abnormal signal can be detected the 1st of March for both Italy and Lombardy, more than one week before the 9th of March. The alarm conditions also hold on the 2nd and the 3rd of March for both Lombardy and Italy. After the 4th of March, the IHR1 drops to 2.274 for Italy and to 3.524 for Lombardy.

To assess the proposed method, we tested it using data from all the Italian regions to check whether the alarm signals are associated with high mortality rates. The alarm signals fire only at the beginning of the first wave of the pandemic; in the other waves, the conditions on HPR1 and IHR1 do not hold simultaneously. Thus, to asses the method, we considered the COVID-19 mortality rate in the first wave only until the 30th of June 2020 [38]. Table 1 summarises the results, which appear promising because the alarm condition only triggers for the nine regions with a higher mortality rate and for the entirety of Italy.

However, recent studies [39,40] indicate that the death cases reported in the official data of the Protezione Civile department (DPC) [38] are underestimated. Considering excess mortality computed using data provided by the National Statistical Institute (ISTAT) by subtracting the average deaths counted in February–April over the prior 5 years (2015–2019) from the deaths counted in the same period in 2020 [29], the results are similar. Indeed, our method identifies the eight regions with higher mortality rates, with the exception that data for Trentino and Alto Adige are aggregated.

Instead, if we look at the estimations of the maxima curves obtained taking the largest between DPC and ISTAT data in each day [29], Friuli Venezia Giulia, which is not selected by our method, goes beyond Veneto. We hypothesize that this issue is due to lack of precision in the functional estimation method used in [29], because the DPC ranking the authors present for the period of February–April 2020 differs from the COVID-19 mortality rates computed at the end of the first wave, and at the end of April 2020, too [38].

## 4. Discussion

Beyond the seemingly good results, a thorough analysis is required. As a first point, we observe that what likely happens is that the data of Italy are influenced by Lombardy and other regions where the alarm signal also fires at the start. Thus, containment strategies that could have been implemented for the whole of Italy are not necessarily those suggested for Lombardy.

Concerning the accusation of culpable epidemic, no new conclusions can be drawn, because the indicators that compose our method were almost unknown (unused) at the beginning of March 2020. For example, the predictive capacity of test positivity with respect to the trend of admissions in hospital and ICUs [27] and its correlation with excess mortality [29] were theorized only afterward.

The question is then whether the proposed method would be able to improve the COVID-19 response strategy in Italy if these alarms were made available. Certainly, the generated alarm signal alone is not sufficient to support decision making; it is just another piece of information that could be considered together with others, such as the three methods used in CIDARS [37,41,42,43]. However, the addition to these known methods of a specific component for infectious diseases with asymptomatic carriers, which can be used along with other signals, can undoubtedly improve surveillance tools for making the right decision on time.

The rationale of this method comes from a surprise effect, considering that the earliest cases in Italy were detected with a considerable delay (up to 3 weeks) [44]. When the first case was detected in Lombardy, the virus had already spread to some areas, and consequently some indicators had out-of-range values in the initial phase. In other words, the rate of outbreak growth was initially underestimated in Italy and many other countries [1].

Associating these anomalous values with a possible excess of mortality in the near future could have provided valuable information at that time, perhaps enough to suddenly implement a lockdown limited to Lombardia, Emilia-Romagna, Veneto, and Marche in the first days of March 2020, potentially avoiding a large number of deaths. Regarding the other regions, Piemonte could have been added a few days later, while for the remaining regions, additional considerations would have to be made. Indeed, the data for Liguria, Trentino, and Valle d’Aosta would probably be affected by an anticipated lockdown in the neighboring regions due to mobility issues [29].

Considering the state of the art of early warning systems in the detection of infectious disease outbreaks, they can be classified according to the source of data collection [3]. The proposed method shares the limitations of all approaches based on public health records that rely on cases reported by clinicians. Therefore, they generally do not work for diseases that are still unknown [37]. Indeed, to use test positivity, the disease should be known, as should the diagnostic test available.

If we compare our approach with other systems and methods based on public health records [37,41,42,43,45,46,47,48,49] designed for monitoring various diseases such as Dengue, Influenza, and COVID-19, all of them rely on known cases and do not utilize test positivity. Consequently, they do not account for the potential spread of asymptomatic cases. Addressing this issue is the most important novelty of our approach.

## 5. Conclusions

Analysing pandemic data in Italy in the first 10 days of the outbreak, we identified anomalous conditions in TPR and in two novel community-level indicators. These conditions allowed for us to define an early warning method capable of signalling a risk for excess mortality in a few days. This method, specific for infectious diseases with asymptomatic carries, can be considered in the implementation of next-generation early warning systems. Using this method, a lockdown limited to Lombardia, Emilia-Romagna, Veneto and Marche could have been implemented in the first days of March 2020. However, it is clear that this knowledge was not available at the beginning of March 2020; thus, nothing new can be concluded concerning the reported accusation of culpable epidemic.

Beyond these seemingly positive results, the generality of the approach requires further investigations. Presumably, it could apply to other respiratory diseases with asymptomatic carriers under a similar criticality COVID-19 condition. Future work will concern a more extensive evaluation, considering data from other countries and diseases. Although the presented method only requires collecting data on administered diagnostic tests and occupancy of beds in hospitals and ICUs, data availability for this assessment remains a critical issue in the initial phase of the pandemic. We also would like to test whether values which exceed lower thresholds for TPR, HPR1 and IHR1 indicators are associated with other critical points in the pandemic and correlate with high mortality rates, investigating variability within COVID-19 waves [50].

## Figures and Tables

**Figure 1 healthcare-12-00469-f001:**
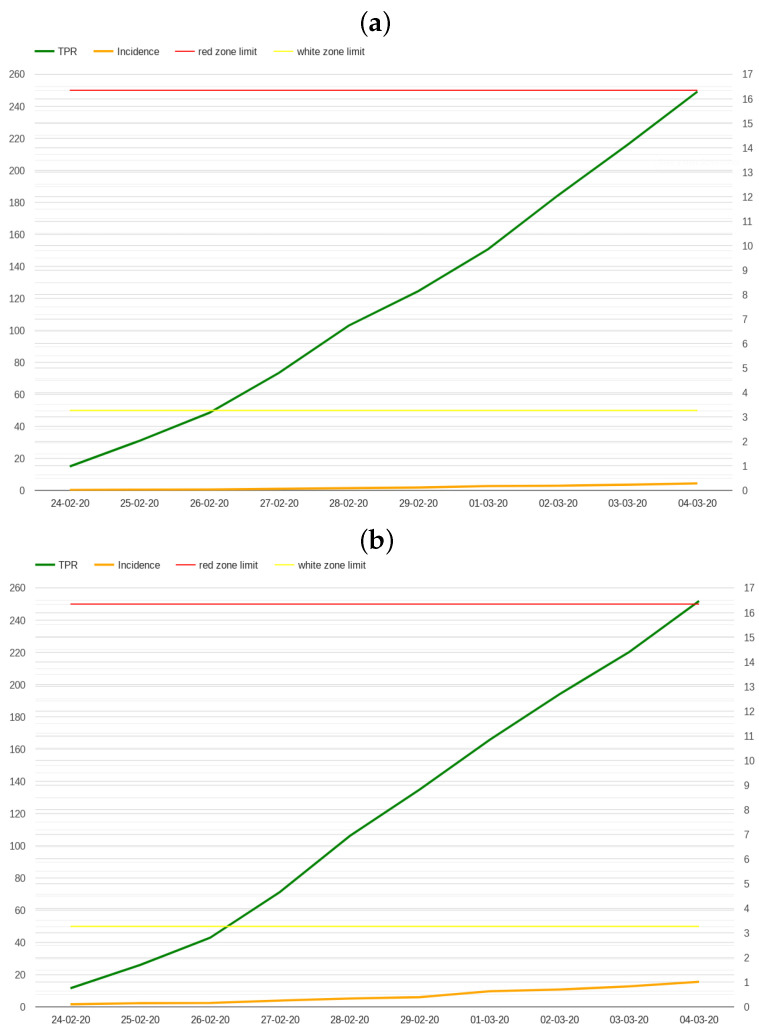
First 10 days of the pandemic in Italy (**a**) and Lombardy (**b**) incidence vs. TPR. The scale on the left indicates incidence rate values, showing the “white zone” and “red zone” thresholds; the scale on the right indicates TPR percentages.

**Figure 2 healthcare-12-00469-f002:**
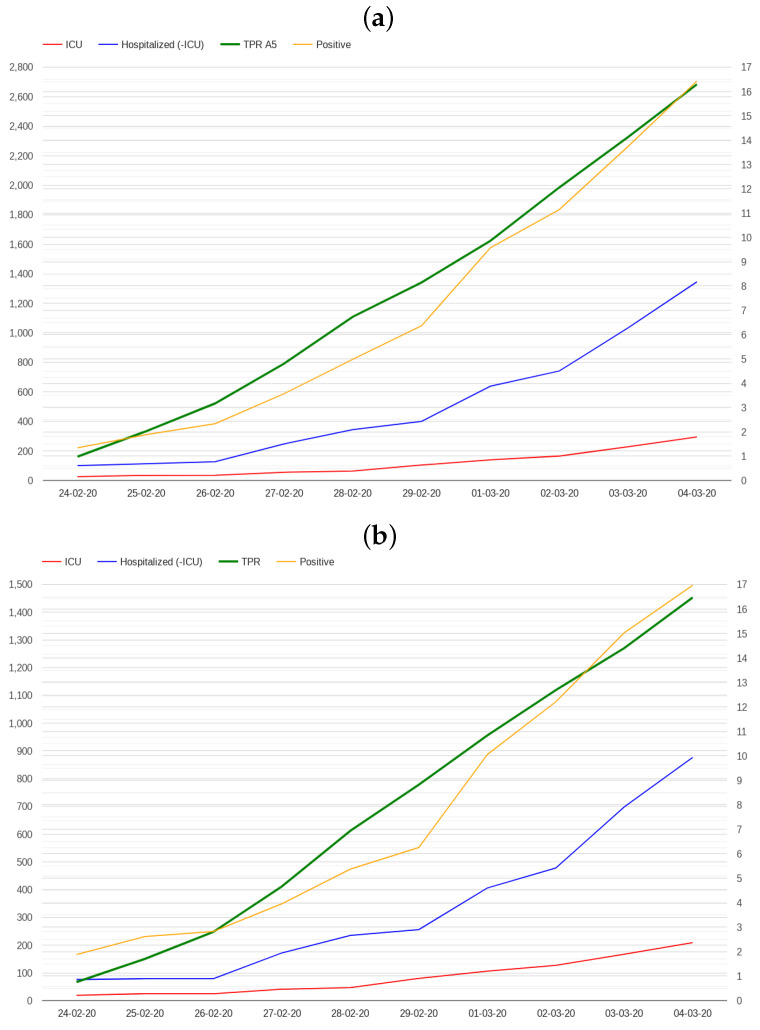
First 10 days of the pandemic in Italy (**a**) and Lombardy (**b**): number of positive cases, number of patients admitted in hospitals and ICUs. The scale on the right indicates TPR percentages.

**Figure 3 healthcare-12-00469-f003:**
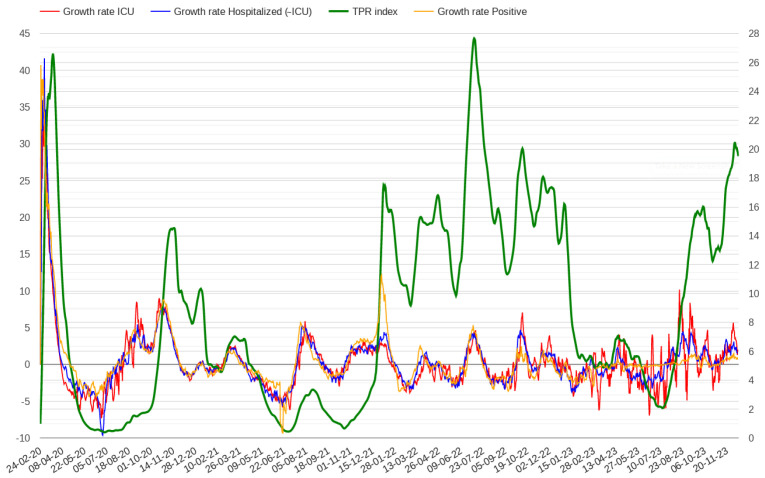
Growth rate of positive cases and admissions in hospitals and ICUs throughout the pandemic in Italy. The scale on the right indicates TPR percentages.

**Figure 4 healthcare-12-00469-f004:**
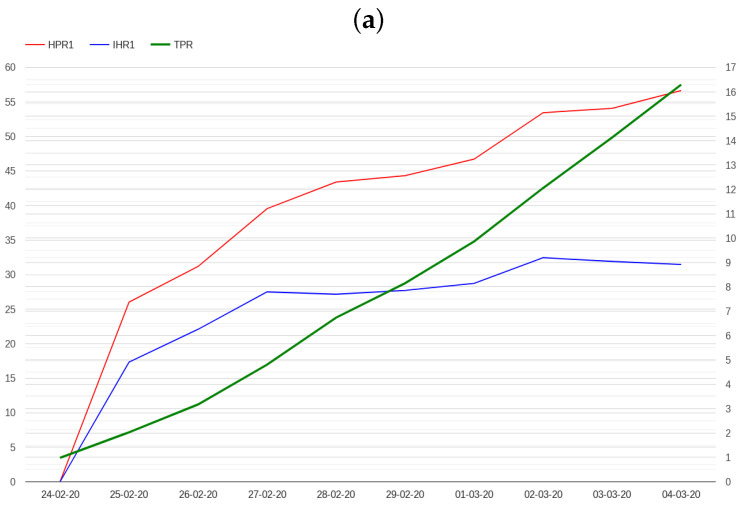
The trend of HPR1 and IHR1 indicators in the first 10 days of the pandemic in Italy (**a**) and Lombardy (**b**). The scale on the right indicates TPR percentages.

**Figure 5 healthcare-12-00469-f005:**
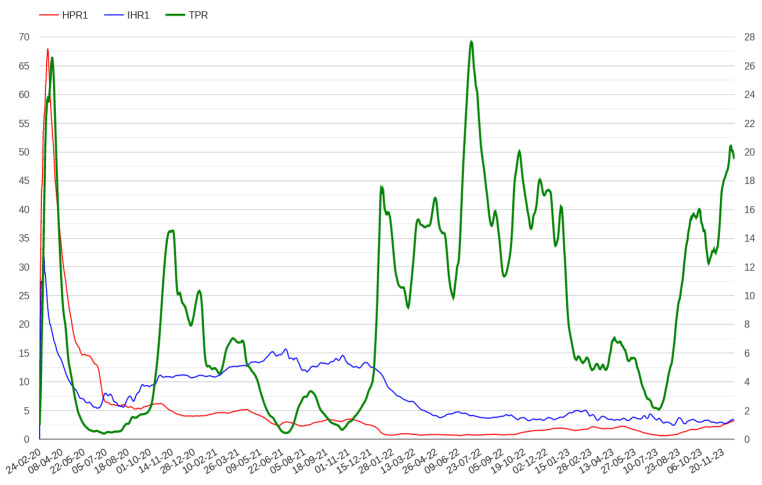
The trend of HPR1 and IHR1 indicators throughout the pandemic in Italy. The scale on the right indicates TPR percentages.

**Table 1 healthcare-12-00469-t001:** This table presents the mortality rates and case fatality rates of all the Italian regions, the alarm signal, the rule that triggers it, and the date of the alarm.

Region	Mortality	Case Fatality	Alarm	Rule	Date
Lombardia	166.141	0.177	1	1	1 March 2020
Valle d’Aosta	116.751	0.122	1	2	20 March 2020
Liguria	101.525	0.156	1	1	12 March 2020
Emilia-Romagna	98.652	0.155	1	1	2 March 2020
Piemonte	94.766	0.13	1	2	5 March 2020
Trentino	75.723	0.09	1	1	12 March 2020
Marche	65.275	0.146	1	1	1 March 2020
ITA	58.172	0.145	1	1	1 March 2020
Alto Adige	55.787	0.111	1	2	15 March 2020
Veneto	41.293	0.104	1	1	2 March 2020
Abruzzo	35.505	0.141	0	-	-
Toscana	29.779	0.108	0	-	-
Friuli Venezia Giulia	28.647	0.104	0	-	-
Lazio	14.324	0.103	0	-	-
Puglia	13.566	0.12	0	-	-
Umbria	9.109	0.056	0	-	-
Sardegna	8.143	0.096	0	-	-
Molise	7.597	0.052	0	-	-
Campania	7.456	0.088	0	-	-
Sicilia	5.664	0.081	0	-	-
Calabria	5.013	0.081	0	-	-
Basilicata	4.826	0.064	0	-	-

## Data Availability

The data used in this paper are made available by the Italian Civil Protection Department and publicly accessible, free of charge, at the following web address: https://github.com/pcm-dpc (accessed on 23 December 2023). The TPR time series are avaliable at the following web address: http://www.cs.unibo.it/~gaspari/www/italy.html (accessed on 23 December 2023).

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
