# Peer review of "A Low-Cost Early Warning Method for Infectious Diseases with Asymptomatic Carriers"

_healthcare, 2024, doi:10.3390/healthcare12040469_

Round 1

Reviewer 1 Report

Comments and Suggestions for Authors

The author investigates Covid-19 data for Italy and specifically for Lombardy region thereof and proposes a new early warning indicator system that would have alarmed the authorities of the onset of the pandemic much earlier. The proposed indicator system performed well against the actual Covid-19 data collected in Italy. It remains unclear how well this system will perform against a different disease with perhaps a longer incubation (latent) period and altrantive transmission mechanism. I would suggest further testing this indicator system against a few other (computer generated) mock data sets prior to acceptance. Otherwise, the paper is very good.

Comments on the Quality of English Language

Minor English editing is necessary. Also, please correct the typos in data.

Author Response

I would like to thank the reviewer for this comments that contributed to improve the quality of the manuscript.

  • All the typos in the text and data have been corrected.
  • Considering that the time to revise the manuscript was only 10 days. it was impossible for me to provide further testing. On the one hand, simulating a realistic dataset is a complex task that I am not able to implement in a few days. On the other hand, data availability in the initial phase of the pandemic is a critical issue. For example, I found a European database reporting hospital and ICU admissions, but for most countries, data do not start from the very beginning.
  • Thus additional testing will be our future work and I emphasized the above argumentation in the conclusions as follows (see red text):

"Future work will concern a more extensive evaluation, considering data from other countries and diseases. Although the presented method only requires collecting data on administered diagnostic tests and occupancy of beds in hospitals and ICUs, data availability for this assessment remains a critical issue in the initial phase of the pandemic. "

Reviewer 2 Report

Comments and Suggestions for Authors

Report on “A low cost early warning method for infectious diseases with asymptomatic carriers

Ref: Healthcare-2825399

Title: A low-cost early warning method for infectious diseases with asymptomatic carriers

Journal: Healthcare

 In this work, the author presented an analysis of the COVID-19 pandemic data of Italy and Lombardy region in the first 10 days of the pandemic, spanning from the 24th of February 2020 to the 4th of March 2020. The objective is to determine whether the use of early warning indicators could have facilitated the identification of a critical increase in infections. This identification, in turn, would have enabled the timely formulation of strategies for pandemic containment, thereby reducing the number of deaths. Finally, the author proposed a low-cost early warning method for infectious respiratory diseases with asymptomatic carriers.

 In my opinion, the collected data, the proposed low-cost early warning method, and the concluding discussions presented in this paper are interesting and I suggest that the paper be accepted for publication in Healthcare.

 Here some minor revisions:

1.      The author should highlight the novelty of his work when compared with similar strategies applied for other epidemies.  

2.      The graphs’ dimensions are large and can be resized.

3.      The references list is short and could be improved by adding more recent references on the same topic. 

4.      Please carefully read and check the language and typos throughout the manuscript. I notice several grammatical errors. For example:

§  Abstract, line 9: “… the use early…” should be “… the use of early…”

§  Please delete the dot at the end of line 40, page 2.

§  Page 9, line 243 (conclusion). “… these knowledge was…” should be “… these knowledges were…”

In general, it can become a good article of medium interest. Still, it needs to improve its presentation in the aspects of writing. 

Comments on the Quality of English Language

Minor editing of English language required according to thr review report.

Author Response

I would like to thank the  reviewer for his comments that contributed to improve the quality of the manuscript.

I addressed the comments as follows:

1) The novelty of the proposed approach has been highlighted in the discussion session adding more references as follows (see the new text in red in the manuscript):

"Considering the state of the art of early warning systems in the detection of infectious disease outbreaks, they can be classified according to the source of data collection [3]. .......

If we compare our approach with other systems and methods based on public health records [37,41–43,45–49], designed for monitoring various diseases such as Dengue, Influenza, and COVID-19, all of them rely on known cases and do not utilize test positivity. Consequently, they do not account for the potential spread of asymptomatic cases. Addressing this issue is the most important novelty of our approach."

2) Graphs have been resized.

3) Several references have been added mostly in the introduction and in the discussion sections passing from 31 to 50.

4) Typos and errors have been corrected by proofreading all the manuscript.

Reviewer 3 Report

Comments and Suggestions for Authors

The topic presented is interesting and the mathematical/statistical methods are sound. The paper is well-written. The topic of infection transmission prior to symptom onset is very important, especially after the COVID-19 pandemic.

To improve the introduction of the paper I would suggest adding some comments and references to mathematical models of infectious diseases with asymptomatic carriers. Among many of them, see Mathematics 2023, 11, 1092. https://doi.org/10.3390/math11051092, for a discrete-time epidemic model dealing with asymptomatic and symptomatic individuals.

Figure 2 is not clear since it mixes two different time scales. I would suggest splitting (a)-(b) and (c). Idem for Figure 3.

Comments on the Quality of English Language

It's fine.

Author Response

I would like to thank the reviewer for his comments that contributed to improve the quality of the manuscript.

I addressed the comments of the reviewer as follows:

1) The introduction as been improved adding several reference to mathematical models that consider asymptomatic cases (see the red text in the manuscript), I am quoting below the new paragraph:

"A critical aspect of COVID 19 pandemic is the presence of a large number of asymptomatic cases, contributing to the rapid worldwide spread [4,5], presenting numerous challenges. Indeed, transmission due to asymptomatic infections cannot be effectively identified, making it more difficult to contain and control [6–9]. New and complex mathematical models are needed to represent epidemic dynamics due to asymptomatic carriers [10–15], also considering infection transmission prior to symptom onset [16]. Most importantly, early detection of aberrations becomes a crucial point[1,17]."

2) The figures 2 and 3 with different time scales have been split and resized (this was the suggestion of another reviewer).

3) All the typos and errors have been corrected by an additional profreading.

Round 2

Reviewer 1 Report

Comments and Suggestions for Authors

The author adequately addressed my concerns.